# Exploring the Relationship between Trichome and Terpene Chemistry in Chrysanthemum

**DOI:** 10.3390/plants11111410

**Published:** 2022-05-26

**Authors:** Yaqin Guan, Sumei Chen, Fadi Chen, Feng Chen, Yifan Jiang

**Affiliations:** 1Key Laboratory of Landscaping, Ministry of Agriculture and Rural Affairs, Key Laboratory of Biology of Ornamental Plants in East China, National Forestry and Grassland Administration, College of Horticulture, Nanjing Agricultural University, Nanjing 210095, China; guanyaqin0826@163.com (Y.G.); chensm@njau.edu.cn (S.C.); chenfd@njau.edu.cn (F.C.); 2Department of Plant Sciences, University of Tennessee, Knoxville, TN 37996, USA; fengc@utk.edu

**Keywords:** volatile compounds, terpene, glandular trichome, flowerhead, emission, Chrysanthemum

## Abstract

Chrysanthemum is a popular ornamental plant with a long history of cultivation. Both the leaf and flowerhead of Chrysanthemum are known to produce diverse secondary metabolites, particularly terpenoids. Here we aimed to determine the relationship between terpene chemistry and the trichome traits in Chrysanthemum. In our examination of three cultivars of *C. morifilium* and three accessions of *C. indicum*, all plants contained T-shaped trichomes and biseriate peltate glandular trichomes. The biseriate peltate glandular trichome contained two basal cells, two stalk cells, six secondary cells and a subcuticular space, while the non-glandular T-shaped trichome was only composed of stalk cells and elongated cells. Histochemical staining analysis indicated that the biseriate peltate glandular trichome contained terpenoids and lipid oil droplets but not the T-shaped trichome. Next, experiments were performed to determine the relationship between the accumulation and emission of the volatile terpenoids and the density of trichomes on the leaves and flowerheads in all six Chrysanthemum cultivars\accessions. A significant correlation was identified between the monoterpenoid and sesquiterpenoid content and the density of glandular trichomes on the leaves, with the correlation coefficients being 0.88, 0.86 and 0.90, respectively. In contrast, there was no significant correlation between the volatile terpenoid content and the density of T-shaped trichomes on the leaves. In flowerheads, a significant correlation was identified between the emission rate of terpenoids and the number of glandular trichomes on the disc florets, with a correlation coefficient of 0.95. Interestingly, the correlation between the density of glandular trichomes and concentrations of terpenoids was insignificant. In summary, the relationship between trichomes and terpenoid chemistry in Chrysanthemum is clearly established. Such knowledge may be helpful for breeding aromatic Chrysanthemum cultivars by modulating the trichome trait.

## 1. Introduction

Chrysanthemum is one of the most well-known traditional Chinese flowers [1] and has potential medicinal and commercial economic value [2]. Studies have reported that Chrysanthemum species are active producers of secondary metabolites [3]. For the chemical categories, terpenoids constitute the largest class of secondary metabolite identified from Chrysanthemum species [4]. At present, more than 435 terpenoids (mainly monoterpenes and sesquiterpenes) have been identified in 36 Chrysanthemum species [5]. These terpenes were widely found in various organs of Chrysanthemum plants, such as flowers, leaves, stems and roots [6,7,8,9,10]. However, numerous studies have only focused on the isolation and characterization of terpenoids from flowerheads and leaves, which are rich in monoterpenoid and sesquiterpenoid volatile organic compounds (VOCs) [11,12,13]. Flowerheads, fresh flowers and flower buds especially have plenty of volatile terpenes in Chrysanthemum [14].

Terpenoids, as the primary constituents in many plant species, serve in attracting pollinators\seed dispersers and defending against pathogens and herbivores [15,16]. Several studies have described the role of terpenes in chemical defense against herbivores in Chrysanthemum. For instance, aphid infestation induced the production and emission of volatile terpenoids that attracted the natural enemies of aphids [17]. Infection with *Botrytis cinerea* induced the leaves of Chrysanthemum to produce monoterpenes and sesquiterpenes to deter herbivores [18]. Furthermore, volatile terpenoids produced in Chrysanthemum have been found to exhibit inhibitory effects on the growth of bacteria and fungi [19,20,21]. Apart from defending against herbivores and antimicrobial activity, the volatile terpene content in Chrysanthemum essential oils has been used as a spice in food and as a natural preservative in the preparation of pharmaceuticals, which have high medicinal and economic value [22,23].

As the storage tissue of the secondary metabolism, volatile terpenes are mainly produced and stored in specialized secretory structures called glandular trichomes [24,25]. Based on their morphology and secretion ability, trichome are classed as unicellular or multicellular and as glandular or non-glandular [26]. In many species, trichomes are considered as true bio-factories able to produce valuable chemicals. For example, peltate trichomes in mint (*Mentha canadensis* Linnaeus) store valuable essential-oil terpenoids [27]. In *Artemisia annua*, glandular trichomes can biosynthesize artemisinin, and thus the artemisinin content can be improved by increasing glandular trichome density [28]. Type VI glandular trichomes serve as chemical factories for production of toxic and repellent substances against arthropod herbivores and pathogens in tomato [29,30]. In addition to their role in defending plants against the attacks of herbivores and pathogens, these specialized substances in glandular trichomes also act as attractants for pollinators and have vital functions as growth regulators and in absorbing UV radiation [31,32].

Glandular trichomes functioning as the unique storage structures of terpenoids have been reported to be involved in the synthesis of terpenoids in some Chrysanthemum species, but the quantitative relationship between the density of trichomes and volatile terpenoid content in the leaves and flowers has not yet been determined. In this study, the volatile terpene content in three cultivars of *C. morifilium* and three accessions of *C. indicum* was analyzed to explore the correlation between terpene production and trichome density. Furthermore, our observation and identification of the types and functions of specific categories of trichomes may provide direction for the breeding of aromatic Chrysanthemum cultivars by modulating glandular trichome initiation.

## 2. Results

### 2.1. Morphological Observation of Trichomes in C. morifilium *cultivars* and C. indicum *accessions*

Observation by SEM showed that two types of trichomes concealed the adaxial leaf (Figure 1A) and abaxial leaf (Figure 1B) of *C. morifilium* cultivars and *C. indicum* accessions: biseriate peltate glandular trichomes (Figure 1D) and T-shaped trichomes (Figure 1C). Furthermore, the glandular trichomes on the adaxial and abaxial sides of the fifth leaves from *C. morifilium* cultivars and *C. indicum* accessions were observed by using fluorescence microscopy. Red spots represent the glandular trichomes (Figure 1E) and the green glowing image shows a magnified view of the glandular trichomes (Figure 1F).

### 2.2. Structure Observation and Histochemistry of Glandular Trichomes and T-Shaped Trichomes

It can be seen that the biseriate peltate glandular trichomes (Figure 2A) in *C. morifilium* cultivars and *C. indicum* accessions contained two basal cells, two stalk cells, six secondary cells and a subcuticular space (Figure 2C,F). Unlike the glandular trichomes, the T-shaped trichomes were composed of stalk cells and elongated cells, which extended in T shapes (Figure 2B). The morphologies of the T-shaped trichomes and glandular trichomes were identical in the six different *C. morifilium* cultivars and *C. indicum* accessions.

To identify whether the Chrysanthemum trichomes could secret and store volatile terpenoids and lipid compounds, we stained frozen sections of leaves with Nadi and Sudan red IV separately. The results showed that sections of Chrysanthemum leaves with Nadi demonstrated a violet gas band in the subcuticular space of the glandular trichomes, indicating the presence of volatile terpenoids, while the T-shaped trichomes were colorless without staining (Figure 2D). Additionally, the subcuticular space of the glandular trichomes showed several lipid oil droplets under the Sudan red IV staining (Figure 2E). Thus, the glandular trichomes were the only trichome type to secrete and accumulate terpenoids and lipids.

### 2.3. Correlation of the Trichome Density and Volatile Terpenoid Content from Leaves of C. morifilium *cultivars* and C. indicum *accessions*

The results from the stereoscopic fluorescence microscopy showed significant variation in the trichome density among these cultivars\accessions. They showed that ‘GJB’ had no glandular type on the adaxial leaves, but the density of glandular trichomes on the whole leaf was equivalent to the other two tea Chrysanthemum cultivars ‘CJ’ and ‘HBJ’ (Figure 3A). Moreover, there were no significant differences in glandular trichome densities on the leaves among the three tea Chrysanthemum cultivars. For the wild Chrysanthemum accessions, the density of glandular trichomes was significantly greater than for the tea Chrysanthemum cultivars. There was striking variation among different species, with ‘LSYJ’ having the highest density. In addition to the glandular trichomes on the leaf epidermis, it was also covered with T-shaped trichomes. The density of T-shaped trichomes on the leaves was higher than that of glandular trichomes. In these Chrysanthemum cultivars\accessions, the wild Chrysanthemum cultivar ‘LSYJ’ had the highest density of T-shaped trichomes (Figure 3B).

Complex terpenoid profiles, including monoterpenoids and sesquiterpenoids, were determined from the leaves from six Chrysanthemum cultivars\accessions (Figure 3C). ‘LSYJ’ was identified as having the highest terpenoid content, followed by ‘NJYJ’ and ‘GJB’, which had the highest terpenoid content among the tea Chrysanthemum cultivars. The concentration of terpenoids in the wild Chrysanthemum ‘LSYJ’ was more than twice that of the tea Chrysanthemum ‘GJB’. It was found that volatile terpenoids from the leaves of wild Chrysanthemum accessions, especially ‘LSYJ’ and ‘NJYJ’, were significantly higher than those from tea Chrysanthemum cultivars. A total of 55 volatile terpene compounds from accumulated concentrations, including monoterpenes, sesquiterpenes and oxygenated terpenes, were identified in the leaves of the six cultivars\accessions (Appendix A). Among these compounds, certain terpenoids accounted for the largest proportion in the six cultivars\accessions. The substance with the highest terpene content in ‘HBJ’ was α-zingeberene (452.68 ± 28.52 µg g^−1^) in the leaves, accounting for 53.31% of the total terpenes content; α-zingeberene was only present in ‘HBJ’ leaves. ‘CJ’ had the highest variety of volatile terpenoids, with the sesquiterpene β-sesquiphellandrene having the highest content (258.15 ± 8.8 µg g^−1^) among them. ‘GJB’ had the highest terpenoid content in tea Chrysanthemum, and the main terpenes were caryophyllene and (*E*)-γ-bisabolene. However, the terpenoid with the highest content from the wild Chrysanthemum ‘LSYJ’ leaves was the monoterpene oxide cis-chrysanthenol (958.8 ± 43.51 µg g^−1^). The terpenes with the highest concentrations in ‘NJYJ’ were the monoterpenes bicyclo [3.1.0] hexane, 4-methylene-1-(1-methylethyl)- and oxide 2-bornanone, accounting for more than 40% of the total terpenes, and the unique cis-β-farnesene (439.45 ± 5.89 µg g^−1^). Although the total terpenoids content in ‘ZJSYJ ’ was the lowest among the six cultivars\accessions, it had the highest β-elemene content among the six Chrysanthemum cultivars\accessions. These results show that the main terpene in the volatile terpene compounds is obviously different among the six cultivars\accessions, which contributes to the specific fragrance.

Trichomes are the main site for the production and storage of secondary metabolites, including volatile terpenes. Combined with the total volatile terpenoid content in the leaves and the trichome density in the six Chrysanthemum cultivars\accessions, the correlation analysis showed that a significant correlation existed between the monoterpenoid, sesquiterpenoid and volatile terpenoid content and the density of glandular trichomes on the leaf, with correlation coefficients of 0.88, 0.86 and 0.90, respectively. We found that there was no significant correlation between the volatile terpenoid content and the density of T-shaped trichomes on the leaf.

### 2.4. Flowerhead Morphology and Glandular Trichome Types of the Six Chrysanthemum cultivars\accessions at the Break Bud Stage

The developmental process of the Chrysanthemum flowerhead can be divided into six stages (tight bud stage, break bud stage, flourishing stage, semi-openness stage, full openness stage and early wilt stage). Since the floral terpenoid content is highest at the break bud stage, the flowerhead at the break bud stage was chosen for the observation of trichome morphology. The observations revealed that the flowerheads of the six Chrysanthemum cultivars\accessions were composed of external whorls of pistillate ligulated ray florets and central whorls of tubular staminate disc florets (Figure 4A). The numbers of ray florets and disc florets per flowerhead from the six Chrysanthemum cultivars\accessions were 19—392 and 54—257, respectively (Figure 5C). The epidermides of the ray florets and disc florets were covered by multicellular biseriate glandular trichomes, which were similar to those previously described for the leaves (Figure 4A). Multicellular biseriate glandular trichomes were present in the abaxial epidermides of ray florets but not in adaxial epidermides (Figure 4B–E). The ray florets and disc florets of the six Chrysanthemum cultivars\accessions had the same type of glandular trichome. The number of glandular trichomes located on the flowerheads of tea Chrysanthemum cultivars was generally higher than for the wild Chrysanthemum accessions (Figure 5D).

### 2.5. The Correlation of Terpenoid Production and Emission and Trichome Density in the Flowerheads of Six Chrysanthemum cultivars\accessions

To analyze the variation in terpenoids in the flowerheads of the six Chrysanthemum cultivars\accessions, the concentration and emission rate of volatile terpenoids were investigated. Results showed that both the content (3406.78 ± 106.26 ug g^−1^) and emission rate (598.67 ± 1.98 ug g^−1^ h^−1^) of volatile terpenoids were highest in the tea Chrysanthemum ‘CJ’ (Figure 5A,B).

A total of 52 volatile terpene compounds, including monoterpenes, sesquiterpenes and oxygenated terpenes, were identified in the flowerheads of the six cultivars\accessions (Appendix A). Among these compounds, certain terpenoids accounted for the largest proportion; for instance, the sesquiterpene β-selinene (676.64 ± 13.15 µg g^−1^) was specifically produced from ‘CJ’ flowerheads, accounting for 31.47% of the total terpene content. The concentration of camphor in the ‘GJB’ flowerhead was 1130.27 ± 67.91 µg g^−1^, accounting for 59.41% of the total terpene content. The concentration of monoterpenes and their oxides in ‘HBJ’ accounting for 70.87% the total terpene content, including eucalyptol, (+)-2-bornanone and cis-chrysanthenol as the main compounds. However, the highest terpenoid content in the flowers of the wild Chrysanthemum species ‘ZJSYJ’ was the oxide cis-chrysanthenol, with a concentration of 945.32 ± 32.82 µg g^−1^. Like ‘ZJSYJ’, cis-chrysanthenol also had the highest concentration in the flowerhead of ‘LSYJ’, with a concentration of 566.52 ± 21.53µg g^−1^. The highest concentration of terpenes in ‘NJYJ’ was oxide 2,6,6-trimethylbicyclo [ 3.2.0 ] hept-2-en-7-one (1318.51 ± 30.55 µg g^−1^).

Compared with wild Chrysanthemum accessions, tea Chrysanthemum cultivars released a great variety of volatile terpenoids, with 3-carene, camphene, eucalyptol, camphor, β-selinene and cis-α-bergamotene as the main compounds (Appendix A). The main emissions of volatile terpenes from wild Chrysanthemum accessions were bicyclo [ 3.1.1 ] hept-2-ene, 3,6,6-trimethyl-, camphene, eucalyptol and camphor (relative values >1% of total amount). Unlike in the leaves, only glandular trichome could be observed (Figure 4B). As the number of ray florets and disc florets per flowerhead varied among the six Chrysanthemum cultivars\accessions, the concentrations and emission rates of terpenoids in the flowerheads were related to the numbers of ray florets and disc florets in the inflorescence. A corresponding analysis was conducted to explore the correlation between the density of glandular trichomes on the ray florets and disc florets and the terpene accumulation and emission. The results showed that there was a significant correlation between the emission rate of terpenoids and the number of glandular trichomes on the disc florets, with a correlation coefficient of 0.95, but there was no correlation with the content (Figure 6).

## 3. Discussion

Chrysanthemum plants synthesize diverse secondary metabolites from leaves and flowers [17,28,33,34]. Trichomes are epidermal protuberances widely present on the surfaces of plants that can assume many different shapes and sizes [31,35]. Two types of trichomes were found on the leaf surfaces in each of the six Chrysanthemum cultivars\accessions (Figure 1C,D), including multicellular T-shaped trichomes (Figure 2B) and biseriate peltate glandular trichomes (Figure 2A,C). The shapes and types of epidermal trichomes were the same as those of *Artemisia annua* [28]. In the histochemical characterization assay, terpenes (Figure 2D,F) and lipid oil droplets (Figure 2E) were observed only in biseriate peltate glandular trichomes, not in T-type trichomes. Previously, terpenes and lipid oil have been demonstrated to be present in glandular trichomes in some Asteraceae and Lamiaceae species [32,36].

In addition to the vegetative organ leaves, the types and distributions of trichomes were also observed in the flowerheads of six Chrysanthemum cultivars\accessions. The typical flowerhead of the Asteraceae family is composed of ray florets and disc florets arranged on a receptacle [37]. We only observed glandular trichomes on the surfaces of the ray florets (Figure 4) and disc florets (Figure 4) at the break bud stage (Figure 4A). This result is consistent with previous studies on Asteraceae species [37,38].

Trichomes on the leaf surface were classified as non-glandular trichomes and glandular trichomes. The non-glandular trichomes generally act as a filter to protect plant tissues against physical damage, while the glandular trichomes may act as chemical defense against external damage [35,39]. The numbers of leaf T-shaped trichome and leaf glandular trichome in wild Chrysanthemum species were generally higher than in tea Chrysanthemum cultivars (Figure 3), which could be related to the high resistance of wild Chrysanthemum accessions. In addition, it was found that the volatile terpene compound content from leaves was significantly correlated with the density of glandular trichomes, but there was no correlation with T-shaped trichome density (Figure 6A). Our results support the hypothesis that glandular trichomes are bio-factories with unique capacities to synthesize, secrete or store many valuable secondary metabolites in plants [35,40].

Floral scents are the primary characteristic of ornamental plants and have a significant influence on plant development/reproduction and human physical/mental health [41,42]. The Asteraceae species is an important ornamental plant with a flowerhead composed of radially arranged ray florets and regular disc florets. Our results showed that there was a significant correlation between the emission rates of the volatile terpene compounds and the number of glandular trichomes in disc florets from the six Chrysanthemum cultivars\accessions (Figure 5B). We hypothesize that the combination of the higher emission of terpenes from the center disc florets and the visual signal color and morphology of marginal ray florets in Chrysanthemum act to attract insects for pollination. It has been reported that disc florets display higher terpenoid emission than ray florets in Chrysanthemum [5]. The petals usually function as tissue for the scattering of the basic parenchyma cells or specialized glandular epidermal cells [43]. In this study, the ray florets and disc florets of six Chrysanthemum cultivars\accessions were both observed to be covered with glandular trichomes (Figure 4B,D). However, there was no significant correlation between the accumulation of volatile terpenoids and the density of glandular trichomes in inflorescences. It is speculated that parenchyma cells were also involved in the synthesis of volatile terpenoids as well as glandular trichomes.

Unlike the glandular trichomes in the leaves, the density of glandular trichomes in the flowerheads of the tea Chrysanthemum cultivars was significantly higher than in the wild Chrysanthemum accessions. By longitudinally comparing the total amounts and types of volatile terpenoids in the flowers and leaves, we found that wild Chrysanthemum accessions had higher concentrations of terpenoids in leaves (Figure 3C), while tea Chrysanthemum cultivars had higher emissions of terpenoids in flowerheads (Figure 5A). These results collectively present a correlation between the accumulation and emission of terpenoids and the density of glandular trichomes on leaves and flowerheads in Chrysanthemum. Furthermore, cis-chrysanthenol, camphor and 2-bornanone were identified as the dominant compounds in the flowerheads and leaves of wild Chrysanthemum accessions. These compounds have been identified as a major terpene component in Chrysanthemum [44] and used as effective plant-derived insect repellents and pesticidal agents [20,45]. However, tea Chrysanthemum cultivars were produced through crossing and artificial breeding with their wild relatives [46]. The capitula of tea Chrysanthemum cultivars possess medicinal and economic value rather than aesthetic characteristics. The flowerheads of tea Chrysanthemum mainly produce and emit 3-carene, camphene, eucalyptol, camphor, 2-bornanone and β-selinene, which have the functions of expelling wind and heat, easing anxiety and improving eyesight [47]. These terpenoids are mainly stored in the glandular trichomes, indicating that the production of volatile terpenes could be increased by increasing the density of glandular trichomes. It has been reported that increasing the density of glandular trichomes in tomato [48], tobacco [49] and sweet wormwood [50] is an effective way to improve the production of specific secondary metabolites. Some transcription factors have been reported to be involved in the regulation of glandular trichomes to increase the production of volatile terpenoids, such as AaHD1 [51], GL1 [52] and AaMIXTA1 [53]. However, whether the regulated initiation of glandular trichomes can be achieved in hexaploid Chrysanthemum remains to be further investigated. Thus, the study of plant trichome development is of great significance for the development of floral fragrance breeding, which needs to be further explored. Moreover, on the basis of the characterization of β-pinene, camphor, α-zingeberene, β-selinene, cis-chrysanthenol and 2-bornanone as the dominant and active terpenoid compounds produced in glandular trichomes of Chrysanthemum, it will also be important to functionally characterize the terpene synthase responsible for the diversity of terpenoids synthesized in glandular trichomes through multi-omics.

## 4. Materials and Methods

### 4.1. Plant Materials

Plants of three cultivars of C. *morifolium* (‘Chu Ju’(CJ), ‘Gong Ju Bai’(GJB(, ‘Hang Bai Ju’(HBJ)) and three accessions of C. *indicum* (‘Nan Jing Ye Ju’(NJYJ), ‘Li Shui Ye Ju’(LSYJ), ‘Zi Jin Shan Ye Ju’(ZJSYJ)) were cultivated in the Chrysanthemum Germplasm Resource Preservation Center, Nanjing Agricultural University (Nanjing Agricultural University, Nanjing, China), with a photoperiod of 16 h light/8h dark at 25 °C and 65% RH (relatively humidity). When the plants grew to 15–20 cm, the fifth leaves below the merism and the flowerhead at the break bud stage were collected from the six cultivars\accessions for the volatile terpenoid identification.

### 4.2. Organic Extraction of the VOCs

Fresh leaf or flowerhead tissue (0.2 g) was ground into powder in liquid nitrogen. Ethyl acetate (Macklin Technology, Shanghai, China) was added into the powder in a 5:1 (volume to weight) ratio, with nonyl acetate (CAS: 143-13-5, ≥98%, Sigma-Aldrich, Saint Louis, MO, USA) included (0.002%) as an internal standard. After shaking at 200 rpm at room temperature for two hours and centrifugation at 5000 rpm for five minutes, the organic phase was collected for the following chemical analysis and bioassays.

### 4.3. Headspace Collection of Volatile Terpenoids from Flowerheads

Ten flowerheads from each Chrysanthemum species/cultivar were placed into an open headspace sampling system (Analytical Research Systems, Gainesvile, FL, USA), as previously described [54]. The dynamic headspace collection was conducted by pumping purified air at a flow rate of 0.7 L/min. VOCs was collected through PorparkQ volatile collection traps (supplied by Analytical Research System, Gainesvile, FL, USA), and volatiles were eluted with 100 µL CH_2_Cl_2_ (Macklin, Shanghai, China) containing nonyl acetate (0.01% *v/v*) as internal standard for quantification. One microliter of eluent was injected into a gas chromatography-mass spectrometry (GC-MS) system for separation and identification of terpenes.

### 4.4. Identification of Volatiles by GC-MS

Terpenoid compounds were analyzed using a GC-MS system (Agilent Intuvo 9000 GC system coupled with an Agilent 7000 Triple Quadrupole mass detector), equipped with an Agilent HP 5 MS capillary column (30 nm × 0.25 mm); helium (99.99%) was used as the carrier gas at a flow rate of 1 mL/min. The injection volume of each sample was 1 µL. The temperature of the injection port was 260 °C, with a spilt mode (spilt ratio = 5:1) at the rate of 5 °C/min. The column oven temperature program was as follows: the temperature was initiated at 40 °C, followed by an increase to 250 °C at a rate of 5 °C/min. The MS condition included an EI ion source temperature of 230 °C, ionization energy of 70 eV and mass scan range of 40–500 amu. The separated constituents were identified using the NIST17 MS library (National Institute of Standards and Technology). A C8-C20 hydrocarbon standard (Sigma-Aldrich, St. Louis, MO, USA) was used to obtain the retention indices. Each constituent was quantified based on the comparison of its peak area with that of the internal standard (Nonyl acetate, Sigma-Aldrich, St. Louis, MO, USA), and the contents were expressed as µg g^−1^ fresh weight.

### 4.5. Scanning Electron Microscopy (SEM) Observation

The fifth leaves were collected for the scanning electron microscopy (SEM) assay and treated following the approach reported previously by [55]. Images were obtained via SEM at 20 Kv (Hitachi S-3000N, Tokyo, Japan).

### 4.6. Glandular Trichome Density Measurement

The fifth leaf and break bud stage capitula were collected to measure the density of glandular trichomes. The assay was performed as described previously [28]. Briefly, images of three parts (the top, middle and bottom) on each side of the leaves and tubular and lingual flowers were obtained using fluorescence microscopy (Leica M165FC, Wetzlar, Germany) with a 5× objective and excitation at 450–480 nm. Trichome number counting was performed using Image J software (https://imagej.nih.gov/ij/download.html).

### 4.7. Histochemical Characterization

Histochemical analyses were performed on 10 μm frozen sections of fresh leaves (Leica CM1950, Wetzlar, Germany). The terpenoids and lipids that existed in glandular trichomes were investigated using the Nadi reaction [56] and Sudan IV [57], respectively. Then, the stained trichomes were observed with a light microscope (Leica DM 6B, Wetzlar, Germany) and images were captured.

### 4.8. Paraffin Section

Fresh samples of the fifth leaves were fixed in FAA 50 [58]. Subsequently, they were subjected to a vacuum to remove the air from the tissues and dehydrated in a graded ethanol series. A fraction of each material was embedded in Lecia Historesin (R) plastic resin (Heraeus Kulzer, Hanau, Germany), and the blocks were sectioned by means of a Lecia RM 2245 rotary microtome at 6 μm. The sections were clarified in 20% sodium hypochlorite, washed in distilled water, stained with safranin and fast green [59] and mounted in 50% glycerin. The distribution and morphology of glandular and T-shaped trichomes was observed via light microscope.

### 4.9. Statistical Analysis

All experiments were performed in three biological replicates. Following a standard analysis of variance, means were compared using Duncan’s multiple comparison (SPSS 25.0; SPSS, Chicago, IL, USA).

## Figures and Tables

**Figure 1 plants-11-01410-f001:**
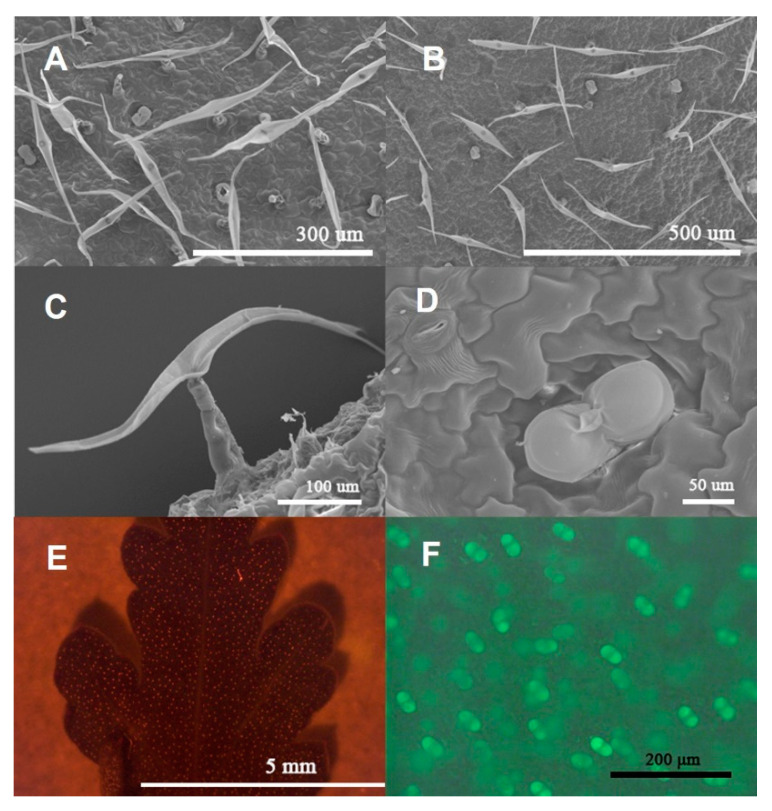
Scanning electron microscopy images of surfaces on mature Chrysanthemum leaves (**A**–**F**). Abaxial leaf of ‘ZJSYJ’ (**A**); adaxial leaf of ‘CJ’ (**B**); T-shape trichomes on the ‘HBJ’ leaf (**C**); glandular trichome on the ‘ZJSYJ’ leaf (**D**); distribution of glandular trichomes on the ‘NJYJ’ leaves under fluorescence microscopy (**E**,**F**).

**Figure 2 plants-11-01410-f002:**
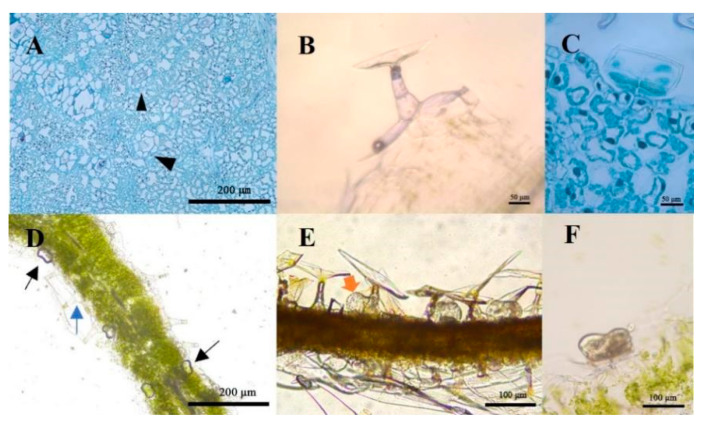
Paraffin section and histochemical staining of the leaf trichomes. Transverse and longitudinal paraffin sections of glandular trichomes (black triangles) in mature leaves (**A**,**C**,**F**) and T-shaped trichomes (**B**). Nadi staining for terpenes revealed that the volatile terpenoids (black arrows) existed in the subcuticular spaces (**D**). The staining for total lipids (yellow arrow) with Sudan IV suggested the accumulation of total lipids in the subcuticular space (**E**).

**Figure 3 plants-11-01410-f003:**
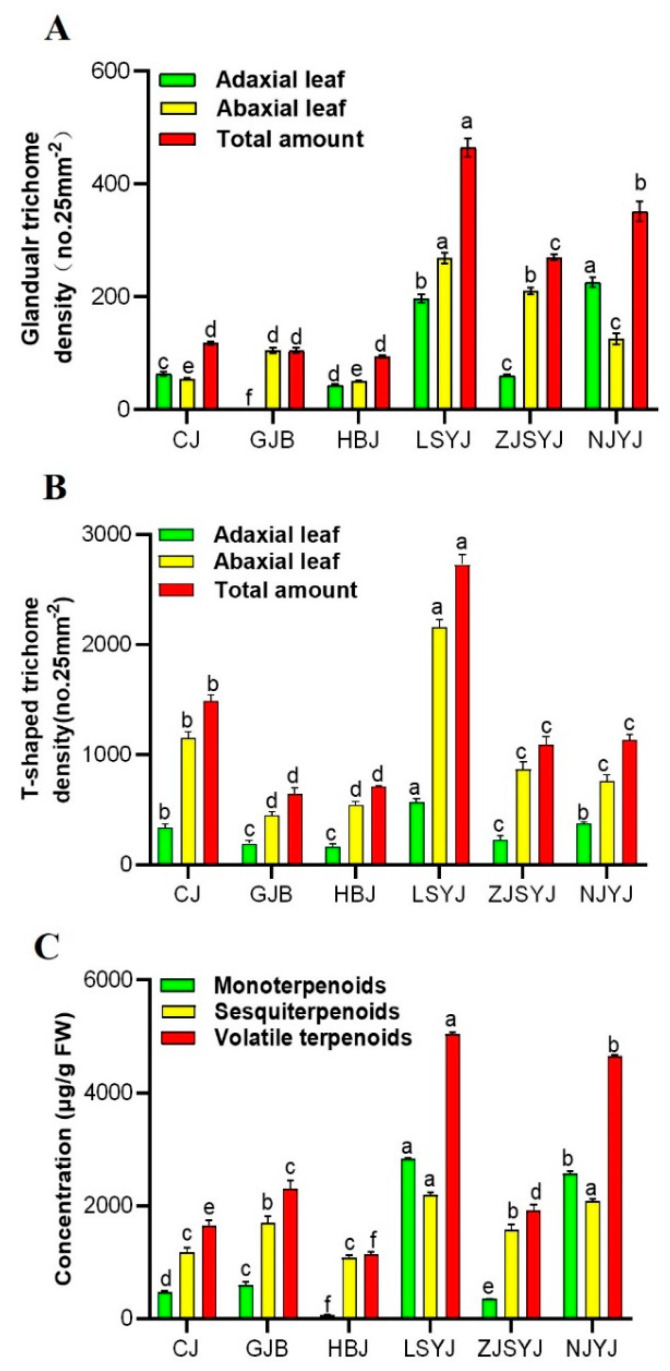
Measurement of trichome density and identification of terpene substances on the leaves of three cultivars of *C. morifilium* (‘CJ’, ‘GJB’ and ‘HBJ’) and three accessions of *C. indicum* (‘LSYJ’, ‘ZJSYJ’ and ‘NJYJ’). Glandular trichome (**A**) and T-shape trichome (**B**) density was measured per 25 mm^2^ on the abaxial surface and adaxial surface of the fifth leaf. The terpenoid compound content in the seventh leaf is also shown (**C**). Data are presented as means ± SD (n = 3). Different letters in (**A**–**C**) denote statistically significant differences among cultivars\accessions according to ANOVA analysis (*p* < 0.05).

**Figure 4 plants-11-01410-f004:**
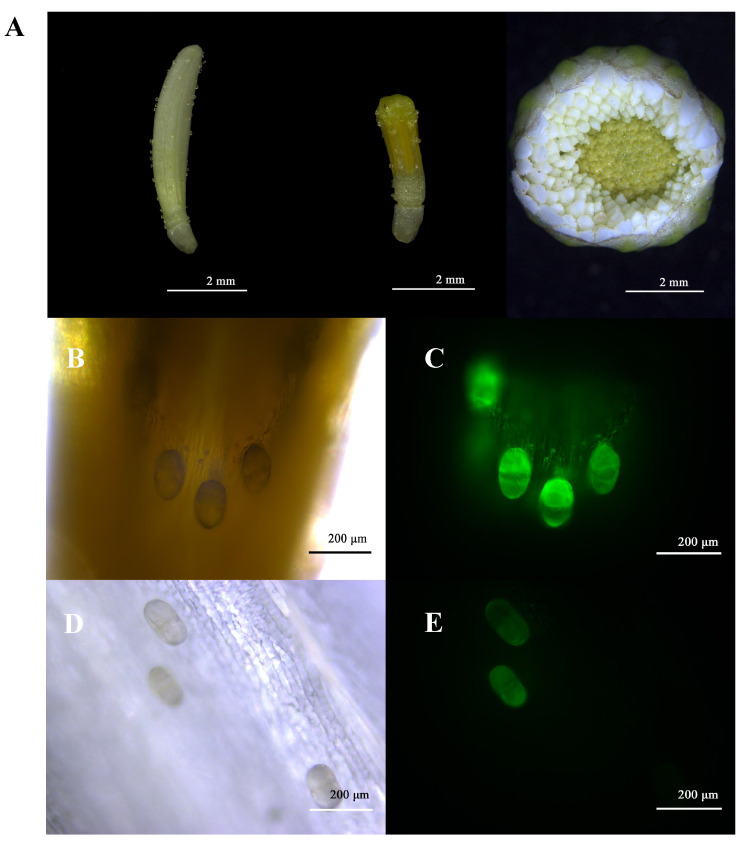
Representative ‘HBJ’ cultivar showing a flowerhead in the break bud stage with an external whorl of ray florets and central whorls of disc florets (**A**); from left to right: ray florets, disc florets and flowerheads. Also shown are trichomes distributed on the ray florets (**B**,**C**) and disc florets (**D**,**E**).

**Figure 5 plants-11-01410-f005:**
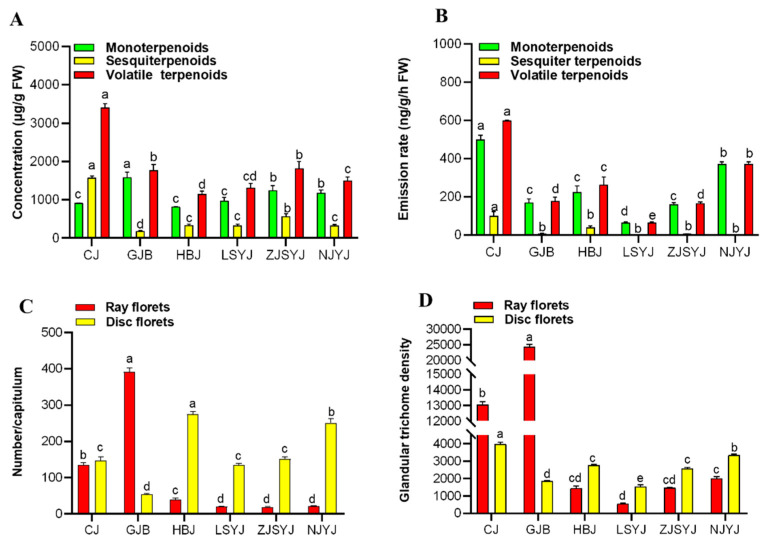
Glandular trichome density and terpene content for the ray florets and disc florets of three cultivars of *C. morifilium* (‘CJ’, ‘GJB’ and ‘HBJ’) and three accessions of *C. indicum* (‘LSYJ’, ‘ZJSYJ’ and ‘NJYJ’): concentration (**A**) and emission rate of terpenoids (**B**); number of ray florets and disc florets per flowerhead (**C**); density of glandular trichomes on ray florets and disc florets (**D**). Different letters in (**A**–**D**) denote statistically significant differences among cultivars\accessions according to ANOVA analysis (*p* < 0.05).

**Figure 6 plants-11-01410-f006:**
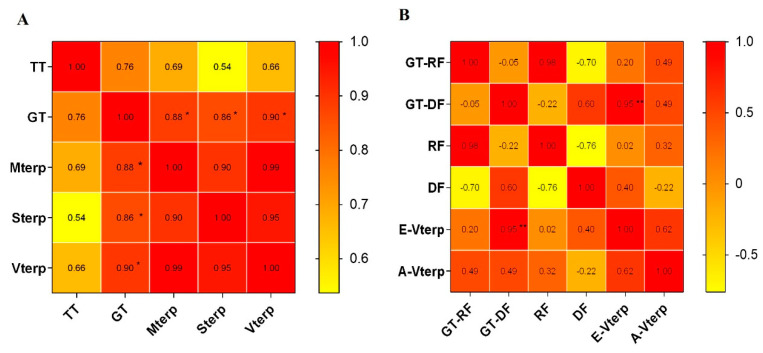
Pearson analysis of the correlation between terpene compounds and the density of glandular trichomes in leaves (**A**) and in flowerheads in the break bud stage (**B**) in six Chrysanthemum cultivars/accessions. TT: T-type trichome; GT: glandular trichome; Mterp: monoterpenoids; Sterp: sesquiterpenoids; Vterp: volatile terpenoids; GT-RF: glandular trichomes of ray florets; GT-DF: glandular trichomes of disc florets; RF: ray florets; DF: disc florets; E-Vterp: emission rate of volatile terpenoids; A-Vterp: accumulation of volatile terpenoids. The color scale ranging from yellow (value,0) to red (value,1) represents the relevant correlation. ** significant at the level of *p* < 0.01; * *p* < 0.05.

## Data Availability

Not applicable.

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
