# Peer review of "Exploring the Relationship between Trichome and Terpene Chemistry in Chrysanthemum"

_plants, 2022, doi:10.3390/plants11111410_

Round 1

Reviewer 1 Report

2022-05-10 Review 

Exploring the relationship between trichome and terpene chemistry in Chrysanthemum

Aim of study:

Guan et al. investigates and quantifies the relationship between the density of trichomes and the content of volatile terpenoids. Besides, a general interest to understand the generation of secondary metabolites many of these are also valuable compounds within e.g medicine. 

By staining the authors are able to show that glandular trichomes is the only trichome type to secrete and accumulate terpenoids and lipids.

The authors show that their results showed that there was an association between the emission rates of volatile terpenes compounds and the number of glandular trichomes.

Overall, it is a neat little story to show an association between glandular trichomes and production of secondary metabolites, especially terpenoids. 

There are some minor English language corrections that would make the paper stronger. I have pointed out a few here below but addressing it thoroughly would benefit the paper. 

Minor typos/language:

L. 33 ..

L. 119 owning -> having

L. 122 owns -> has

Check owning versus have 

Figure 5: spelling mistake monterpenoid -> monoterpenoid

There are some questions around the results obtained and lack of description of measurement that would increase the quality of the paper:

  1. In figure 3 and 5 different letters indicate statistically significant differences - but which is not clear from the figure text and hence hard to understand and interpret? 

  2. Volatile terpenes

    1. Not clear why e.g., alpha-Zingeberene, cis-Chrysanthenol were investigated among many other secondary metabolites? 

    2. Measurements of these compounds are not clearly described or discussed - e.g. how much of the measurement could be explained by the MS? 

Addressing these questions would make the paper better. 

Author Response

Reviewer 1

Aim of study: Guan et al. investigates and quantifies the relationship between the density of trichomes and the content of volatile terpenoids. Besides, a general interest to understand the generation of secondary metabolites many of these are also valuable compounds within e.g medicine. By staining the authors are able to show that glandular trichomes is the only trichome type to secrete and accumulate terpenoids and lipids.The authors show that their results showed that there was an association between the emission rates of volatile terpenes compounds and the number of glandular trichomes. Overall, it is a neat little story to show an association between glandular trichomes and production of secondary metabolites, especially terpenoids. There are some minor English language corrections that would make the paper stronger. I have pointed out a few here below but addressing it thoroughly would benefit the paper. 

Thanks for the positive comments of our manuscript.

(1): Minor typos/language: “L. 33 ..”, “L. 119 owning -> having”, “L. 122 owns -> has”, “Check owning versus have”, “Figure 5: spelling mistake monterpenoid -> monoterpenoid”.

Response: We have revised and corrected these mistakes as suggested.

(2): There are some questions around the results obtained and lack of description of measurement that would increase the quality of the paper:  In figure 3 and 5 different letters indicate statistically significant differences - but which is not clear from the figure text and hence hard to understand and interpret? 

Response: We have added the interpretation of the different letters indicating the significant differences in the figure legend as suggested.

(3): Not clear why e.g., alpha-Zingeberene, cis-Chrysanthenol were investigated among many other secondary metabolites? Measurements of these compounds are not clearly described or discussed - e.g. how much of the measurement could be explained by the MS?

 Response: Thanks for the critical comments. We highlight the discussion of alpha-Zingeberene and cis-Chrysanthenol, because these two compounds were identified as the dominant compounds in wild Chrysanthemum accessions and showed relation with the density of the trichome.

Reviewer 2 Report

Manuscript can be accepted

Author Response

Thanks for the decision.

Reviewer 3 Report

I found the manuscript "Exploring the relationship between trichome and terpene chemistry in Chrysanthemum" prepared by Guan et al. to be interesting and appropriate to the scope of the Plants journal. I don't have a significant remark on this manuscript. Small advice before final decision. 

line 38: word "famous" can be deleted.  "cultivars\accessions" too often used in the text, Authors should re-arranged this phrase.

Sometimes "volatilized" could be used instead of "volatile". Abbreviation VOCs as volatile organic compounds could be incorporated into the text.

Line 63: should be [24, 25].

what database was used to identify volatile compounds? Lack of such info in subchapter: "Identification of Volatiles by GC-MS".

Fig 5. B probably there is a mistake in the compound's name. There is no table with separated volatile compounds that were determined in the material and mentioned in the text (2.3 and 2.5 subchapters).

Names of compounds such as: 3-Carene, Camphene, Eucalyptol, Camphor, 2-Bornanone and β-Selinene could be written by small letter.

Authors should check if references are prepared according to journal recommendations.

Author Response

Reviewer 3

I found the manuscript "Exploring the relationship between trichome and terpene chemistry in Chrysanthemum" prepared by Guan et al. to be interesting and appropriate to the scope of the Plants journal. I don't have a significant remark on this manuscript. Small advice before final decision. 

Thanks for the positive comments of our manuscript.

(1): line 38: word "famous" can be deleted. "cultivars\accessions" too often used in the text, Authors should re-arranged this phrase.

   Response: We have restated these phrases as suggested.

(2): Sometimes "volatilized" could be used instead of "volatile". Abbreviation VOCs as volatile organic compounds could be incorporated into the text.

Response: We replaced ‘volatile’ with "volatilized" and incorporated VOCs as  suggested.

(3): Line 63: should be [24, 25].

Response: We have revised it as suggested.

(4): what database was used to identify volatile compounds? Lack of such info in subchapter: "Identification of Volatiles by GC-MS". 

Response: We have added more information about the database into the section "Identification of Volatiles by GC-MS" as suggested.

(5): Fig 5. B probably there is a mistake in the compound's name. There is no table with separated volatile compounds that were determined in the material and mentioned in the text (2.3 and 2.5 subchapters).

 Response: Thanks for the critical comments. This table listing the individual compounds was supplied as the supplementary materials (Table S1 and Table S2).

(6): Names of compounds such as: 3-Carene, Camphene, Eucalyptol, Camphor, 2-Bornanone and β-Selinene could be written by small letter.

    Response: We have revised it as suggested.

(7): Authors should check if references are prepared according to journal recommendations

   Response: We have checked the format of all references as suggested.